Isolation, identification, and pathological effects of beach sand bacterial extract on human skin keratinocytes in vitro

Subhan Fazli 1
Shahzad Raheem 2
Tauseef Isfahan 1
Haleem Kashif Syed 1
Rehman Atta-Ur 3
Mahmood Sajid 3
Lee In-Jung ijlee@knu.ac.kr 2 4
1 Department of Microbiology, Hazara University , Mansehra , Pakistan
2 School of Applied Biosciences, Kyungpook National University , Daegu , Republic of Korea
3 Department of Zoology, Hazara University , Mansehra , Pakistan
4 Research Institute for Dok-do and Ulleung-do Island, Kyungpook National University , Daegu , Republic of Korea
Tulkens Paul
Electronic publication date: 2018 Jan 12
Publication date: 2018
Volume: 6
Electronic Location ID: e4245
Received 2017 Sep 27; Accepted 2017 Dec 18
Copyright: ©2018 Subhan et al.
Copyright year: 2018
Copyright holder: Subhan et al.
License: This is an open access article distributed under the terms of the Creative Commons Attribution License, which permits unrestricted use, distribution, reproduction and adaptation in any medium and for any purpose provided that it is properly attributed. For attribution, the original author(s), title, publication source (PeerJ) and either DOI or URL of the article must be cited.
License URL: https://creativecommons.org/licenses/by/4.0/

Keywords: Sequencing, Skin, Pathology, Bacteria, Beach sand

Funding: Basic Science Research Program through the National Research Foundation of Korea (NRF) Ministry of Education 2016R1A6A1A05011910 This research was supported by the Basic Science Research Program through the National Research Foundation of Korea (NRF) funded by the Ministry of Education (2016R1A6A1A05011910). The funders had no role in study design, data collection and analysis, decision to publish, or preparation of the manuscript.

==============================
Background

Beaches are recreational spots for people. However, beach sand contains harmful microbes that affect human health, and there are no established methods for either sampling and identifying beach-borne pathogens or managing the quality of beach sand.

Method

This study was conducted with the aim of improving human safety at beaches and augmenting the quality of the beach experience. Beach sand was used as a resource to isolate bacteria due to its distinctive features and the biodiversity of the beach sand biota. A selected bacterial isolate termed FSRS was identified as Pseudomonas stutzeri using 16S rRNA sequencing and phylogenetic analysis, and the sequence was deposited in the NCBI GenBank database under the accession number MF599548. The isolated P. stutzeri bacterium was cultured in Luria–Bertani growth medium, and a crude extract was prepared using ethyl acetate to examine the potential pathogenic effect of P. stutzeri on human skin. A human skin keratinocyte cell line (HaCaT) was used to assess cell adhesion, cell viability, and cell proliferation using a morphological analysis and a WST-1 assay.

Result

The crude P. stutzeri extract inhibited cell adhesion and decreased cell viability in HaCaT cells. We concluded that the crude extract of P. stutzeri FSRS had a strong pathological effect on human skin cells.

Discussion

Beach visitors frequently get skin infections, but the exact cause of the infections is yet to be determined. The beach sand bacterium P. stutzeri may, therefore, be responsible for some of the dermatological problems experienced by people visiting the beach.

Introduction

Beaches are recreational sites where numerous leisure activities can be carried out, including sunbathing and various sports and games. Therefore, beaches are economically important, and the care and maintenance of the coastal environment is important in many countries (Halliday & Gast, 2011).

Unfortunately, there have been several negative reports regarding the condition of beaches and the incidence of diseases acquired by visitors. For example, Heaney and colleagues reported that visitors to both freshwater and marine beaches in different parts of the USA in 2003–2005 had an increased chance of contracting diarrhea, vomiting, and nausea. Staphylococcus saprophyticus is a causative agent of skin and urinary tract infections in young women, and this bacterium was reported to be present at a Brazilian beach (De Sousa et al., 2017). There have been several other reports of similar issues, including reports by the Environmental Protection Agency and the Centers for Disease Control and Prevention (Spivey, 2009) and a recent paper (He et al., 2017).

The skin is the largest organ in the body and provides a primary barrier against various infectious agents. Because the skin is constantly in direct contact with the external environment, it harbors multiple types of microbes. Some of those microbes are potentially pathogenic and could lead to serious infections under certain conditions (Soufi & Soufi, 2016). The most common pathogenic bacteria in the seaside environment belong to the genera Staphylococcus and Streptococcus (Lee et al., 2016). Similarly, the most common bacteria that colonize the skin are usually gram-positive species, including Staphylococcus epidermidis, Corynebacterium species, Staphylococcus aureus, and Streptococcus pyogenes (Todar, 2008; Braun et al., 2015). Human skin is one of the most significant targets for the pathogenic bacteria present at beaches. In this regard, it has been reported that about 24% of the UK civilian population experiences a pathological skin condition each year. In 2009–2010, about 900,000 patients were referred to dermatologists in England (Tanzer, Macdonald & Schofield, 2014).

A healthy and functioning skin barrier provides protection against dehydration, the penetration of various microorganisms, allergens, chemical irritants, reactive oxygen species, and radiation. Human keratinocytes are the key cells in the skin barrier that provide the skin’s protective function (Salmon, Armstrong & Ansel, 1994). The skin has several physical defense mechanisms that allow it to protect the body against pathogenic microbial infections, including a low pH and the secretion of sebaceous fluid and fatty acids, which act together to inhibit pathogen growth. In addition, the skin possesses its own normal protective flora (McAdam, 2005).

Certain less common bacterial skin infections, such as those by Vibrio vulnificus, are acquired in hospitals or nursing homes, and after swimming in a pond, lake, or ocean. Vibrio vulnificus is a pathogenic gram-negative bacterium that is motile, curved, and rod-shaped (Jones & Oliver, 2009). V. vulnificus infections have been reported in swimmers with open wounds, and they can cause ulceration of the skin (Horseman & Surani, 2011). The increase in the frequency of beach-visits by people has led to an increased incidence of bacterial infections. Therefore, this topic needs to be explored to improve the safety of humans visiting beaches.

Pseudomonas stutzeri is a gram-negative, aerobic, non-fermenting, and oxidase-positive bacterium. P. stutzeri is most commonly isolated patient samples in hospitals (e.g., from surgical wounds, blood, the respiratory tract, and urine) (Holmes, 1986; Taneja et al., 2004). The prevalence of P. stutzeri in settings other than hospitals has not been explored. This study aimed to confirm the presence of P. stutzeri in beach sand and then examine the effects of a P. stutzeri extract on the cell adhesion and cell viability of a human skin keratinocyte to understand whether P. stutzeri presents a potential pathological threat to beach-goers.

Methods

Sample collection and isolation of bacterial strains

Sand samples were collected from the beaches in Busan, Republic of Korea. The contaminating plant debris were removed. Each sample (10 g) was transferred to an autoclaved flask that contained sterile Amies solution (Amies, 1967). The resulting suspension was serially diluted (10−4) and 0.1 mL of the diluted suspension was plated onto Luria–Bertani (LB) agar medium (10 g tryptone, 5 g yeast extract, and 10 g NaCl, pH 7.0 ± 0.2, autoclaved for 15 min at 121 °C) and incubated at 28 °C for 48 h (Kang et al., 2015). The plates were inspected daily. Colonies were distinguished based on their morphology and subsequently streaked onto fresh plates.

Identification and phylogenetic analysis

The taxonomic identification of the selected bacterial isolate, Fazli Subhan Raheem Shahzad (FSRS), was performed based on sequencing its 16S rRNA gene using the 27f and 1492r primers and BLASTing the FSRS 16S rRNA sequence against the NCBI GenBank database, as described by Shahzad et al. (2016). Closely related sequences with the highest homology (i.e., the lowest E values) were selected and aligned by ClustalW using the MEGA version 7.0 software.

Preparation of a crude bacterial extract

An ethyl acetate extract of the FSRS bacterial isolate was prepared by culturing FSRS in LB broth for seven days at 28 °C and 120 rpm in a shaking incubator. After seven days, the culture broth was centrifuged at 10,000× g, and cell-free LB broth was adjusted to pH 2.5 and then completely extracted with an equal volume of ethyl acetate three times. The extracted ethyl acetate was then completely dried using a rotary evaporator to obtain a crude solid bacterial ethyl acetate extract, which was then re-dissolved in sterilized RNase-free water and used for further study.

Preparation of a working solution of the crude bacterial extract

A 5 mg/mL stock solution of the crude extract was prepared using distilled water as the solvent. The solution was filter-sterilized and a working solution of 1 mg/mL was prepared for the final treatments at doses of 100, 300, and 500 ng/mL.

Cell culture

HaCaT keratinocytes were purchased from the Cell Line Service of the German Cancer Research Center (Heidelberg, Germany) and maintained in Dulbecco’s modified Eagle medium (Life Technologies, Carlsbad, CA, USA) supplemented with 10% fetal bovine serum (Life Technologies), 100 U/mL penicillin, and 100 mg/mL streptomycin (Life Technologies) at 37 °C in a humidified atmosphere containing 5% CO2.

HaCaT cell adhesion

HaCaT cells were seeded into six-well plates at a density of 2 × 105 cells/well in 3 mL of medium. The cells were treated with varying concentrations (0, 100, 300, and 500 ng/mL) of crude bacterial extract for 24 and 48 h in serum-free medium. Morphological changes of the HaCaT cells were observed and images were captured under an inverted light microscope (Nikon Corporation, Tokyo, Japan) after 24 and 48 h. The same area of cells was noted and captured at each time point. The images were captured at 200 × magnification and the scale bar represents 50 µm.

HaCaT cell viability

HaCaT cells were plated in flat-bottomed, 96-well, microtiter plates at 1 × 104 cells/well and cultured for 24 h. After starvation in serum-free media for 24 h, the cells were treated with the crude bacterial extract for a further 24 or 48 h. Cell proliferation was determined after treatment using a colorimetric water-soluble tetrazolium salt-1 (WST-1) cell proliferation assay (EZ-Cytox assay kit; Daeil Lab Service, Seoul, Korea). Metabolically active cells generate a colored dye through the action of a cellular dehydrogenase on WST-1, which can be measured using a microplate reader (Tecan, Männedorf, Switzerland) at 450 nm according to the manufacturer’s instructions. The IC50 value were calculated as presented here is given as percent inhibition by the drug. The IC50 were calculated using Curve-Fit Model through Table curve Software and the following formula was used.

The value were subtracted from 100 as total cells-viable cells = percent inhibited cells.

Statistical analysis

The data are expressed as mean ± standard deviation. The results were statistically analyzed by a two-tailed Student’s t-test. Statistically, differences were considered to be significant at p < 0.05.

Figure 1 The geographical location of a beach in Busan, Republic of Korea, which attracts numerous visitors, from where sand samples were collected.

Results

Isolation and identification

To gain insight into the genome sequence diversity of P. stutzeri populations, various isolates of P. stutzeri were selected from beach sands in South Korea (Busan) and several other geographical locations (Fig. 1). Among the isolated strains, the FSRS strain was selected. The 16S rRNA sequence of FSRS was compared to other 16S rRNA sequences in the NCBI database (http://www.ncbi.nlm.nih.gov/) using a BLAST search to identify its closest neighbors. The results showed that FSRS was closely related to members of the genus Pseudomonas and shared the highest sequence identity (100%) with P. stutzeri (KF894696).

A phylogenetic or evolutionary tree is a branching diagram that represents the inferred evolutionary relationships among various biological species based upon similarities and differences in their physical or genetic characteristics. A phylogenetic analysis based on 16S rRNA sequences showed that FSRS is closely related to other P. stutzeri strains, as shown in Fig. 2. A phylogenetic tree was constructed by the neighbor-joining method using CLUSTAL-X 1.8 software (Sasson et al., 2017). Based on this analysis, FSRS was formally identified as P. stutzeri, and its 16S rRNA gene sequence was deposited in the NCBI GenBank database under accession number MF599548.

Figure 2 Neighbor-joining phylogenetic tree derived from aligning the most similar 16S rRNA sequences in related taxa for a phylogenetic analysis of the bacterial isolate (FSRS) by MEGA 7.

The scale bar indicates 0.02 substitutions per nucleotide position. Bootstrap values (%) based on 1,000 replications are given at each branch point.

HaCaT cell adhesion

The effect of FSRS on the adhesion of human skin keratinocytes was determined using HaCaT cells. HaCaT cells were seeded in a six-well plate, treated with different amounts of crude FSRS extract for 24 or 48 h, and then examined by phase contrast microscopy. The results showed that the crude FSRS extract inhibited HaCaT cell adhesion in both a dose- and time-dependent manner, as shown in Fig. 3.

Figure 3 Human skin keratinocyte number and adhesion as assessed by phase contrast microscopy after 24 hr (A) and 48 hr treatment (B).

The number of HaCaT cells was reduced and their cell morphology was changed by the FSRS extract in both a dose- and time-dependent manner, as compared to control cells.

HaCaT cell viability

We determined the effect of the FSRS extract on HaCaT cell viability using a WST-1 assay (Subhan et al., 2017). HaCaT cells were cultured in a 96-well plate and then starved for 24 h before treatment with the FSRS crude extract. As shown in Fig. 4, the crude FSRS extract significantly decreased the viability of HaCaT cells in both a dose- and time-dependent manner. The IC50 value were calculated, which are 198.58 ng/mL for 24 hr treatment and 97.61 ng/mL for 48 hr treatment.

Figure 4 HaCaT cell viability as assessed using a WST-1 assay.

HaCaT cell viability was significantly inhibited by the FSRS crude extract in both a dose- and time-dependent manner, as compared to control cells (n = 3, p < 0.05). The IC50 value were calculated, which are 198.58 ng/mL for 24 hr (A) treatment and 97.61 ng/mL for 48 hr (B) treatment.

Discussion

P. stutzeri was first named by Burri & Stutzer (1895), and later by Van Niel & Allen (1952), while it was phenotypically described by Lehman & Neumann (1896). P. stutzeri has previously been identified in contaminated and non-contaminated soils, marsh sediment, and marine water (Sikorski, Lalucat & Wackernagel, 2005; Lidbury et al., 2016). P. stutzeri is generally considered to be a contaminant that can also be pathogenic to humans if their immunity is compromised. To our knowledge, this is the first study to reveal the existence of pathogenic P. stutzeri in the sand from a beach in Busan, South Korea, and furthermore demonstrate the pathological effects of an extract of P. stutzeri on human skin.

We cultured beach-sand isolates and used the restriction fragment length polymorphism method to identify their 16S rRNA genes. This method has previously been described as being useful for the characterization of bacterial isolates (Adderson et al., 2008). Through that analysis, the isolate was confirmed to belong to the genus Pseudomonas and the species P. stutzeri (Bennasar et al., 1996; Zhang et al., 2014). The 16S rRNA sequence was subsequently submitted to NCBI (KF894696). Several different strains of P. stutzeri have been isolated from different sources, including patient samples in hospitals (wound pus, blood, urine, tracheal aspirates, and sputum), and they are often considered to be pathogenic (Stan, Lim & Sakazaki, 1977; Ergin & Mutlu, 1999; Manfredi et al., 2000). Consistent with this, several other species of the genus Pseudomonas are also considered to be pathogenic, although some of them have applications in bioremediation (Reisler & Blumberg, 1999; Castaldo et al., 2017; Mukherjee et al., 2017).

Having a sufficient understanding of the human pathological microbiome is necessary for the development of successful therapeutic approaches for their cure. Hence, it is important to identify and analyze their possible pathological effects on different parts of the human body, such as skin. In the present study, we demonstrated the presence of P. stutzeri, a potential human pathogen, in sand obtained from a beach in Busan, South Korea.

A previous study has reported the pathogenic effects of P. stutzeri on skin (Puzenat et al., 2004); hence, we focused on examining the effects of P. stutzeri extracts on skin in the present study. We decided to examine the effect of an extract of P. stutzeri on skin keratinocytes because they serve as the first line of defense against foreign invaders in humans. The HaCaT cell line was chosen for this study because HaCaT cells have been widely used as a keratinocyte model cell line in numerous dermatopathological studies.

Skin continually interacts with the external environment. As a result, it is exposed to and colonized by both pathogenic and non-pathogenic bacterial strains. Most of the skin surface-colonizing bacteria are gram-positive species including Staphylococcus aureus and Streptococcus pyogenes (Kanayama et al., 2016). The normal flora in the skin provides protection against various pathogenic microflora, but the exact mechanism of interaction between the pathogenic and normal microflora in the skin is not well understood (Grice & Segre, 2011). It is thought that the normal skin flora secretes a variety of antimicrobial substances that protect human skin from pathogenic bacterial infections (Weinberg, Krisanaprakornkit & Dale, 1998; Rippon, Colegrave & Ousey, 2016).

Figure 5 Schematic representation of beach sand sample collection from the collection site, and the processing, identification, and effect of the FSRS crude extract on the adhesion and viability of HaCaT cells in vitro.

In addition to hosting the normal human skin flora, human skin serves as the first line of defense against pathogenic microbial infection by providing a physical barrier, a low pH, sebaceous fluid, and fatty acids, which together serve to inhibit the growth of pathogens (Peschel et al., 2001). Unfortunately, several pathogenic microorganisms can bypass these defensive mechanisms and penetrate the integument, leading to tissue damage and the induction of an inflammatory response (Roesner, Werfel & Heratizadeh, 2016). Cellulitis is a common bacterial infection of the skin that is caused by Staphylococcus or Streptococcus spp. that are commonly present on the skin (Swartz, 2004; Stevens et al., 2004). Similarly, Pseudomonas spp. can also cause an infection when there is a break in the skin (Nakagami et al., 2011). Currently, it is not understood how bacteria found in beach sand can cause skin infections such as cellulitis. Through this study, we make a new proposal that FSRS in beach sand may cause skin infections by inhibiting keratinocyte adhesion and reducing keratinocyte viability, although further detailed studies are required to fully understand the mechanism of action.

Conclusions

In conclusion, the present work demonstrated that bacterial isolates from beach sand collected from Busan in South Korea contain FSRS, and their identification was based on 16S rRNA sequencing data. An extract prepared from beach sand FSRS significantly inhibited the growth of a human skin keratinocyte cell line by inhibiting cell adhesion and decreasing cell viability (Fig. 5). The extract from FSRS is one of the most potent microbial extracts tested against a human skin keratinocyte cell line. Therefore, this study could be useful for understanding the pathology of skin infections acquired at the beach.

Supplemental Information

Supplemental Information 1 Raw data cell viability after 24 hrs

Click here for additional data file.

Supplemental Information 2 Raw data cell viability after 48 hrs

Click here for additional data file.

Additional Information and Declarations

Competing Interests

Author Contributions

Data Availability

The authors declare there are no competing interests.

Fazli Subhan and Raheem Shahzad performed the experiments, analyzed the data, wrote the paper, prepared figures and/or tables, reviewed drafts of the paper.

Isfahan Tauseef, Kashif Syed Haleem, Atta-Ur Rehman and Sajid Mahmood analyzed the data, reviewed drafts of the paper.

In-Jung Lee conceived and designed the experiments, contributed reagents/materials/analysis tools, reviewed drafts of the paper.

The following information was supplied regarding data availability:

The raw data has been provided in the Supplemental Files.

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
