# Peer review of "Isolation, identification, and pathological effects of beach sand bacterial extract on human skin keratinocytes in vitro"

_PeerJ, doi:10.7717/peerj.4245_

## Round 0.1 · original submission · Major Revisions

As you will see, your paper has been examined by 3 reviewers who made a number of useful suggestions. Although each individual reviewer indicated only "minor revision", the number of suggestions make me willing to examine in details your rebuttal and how the paper has been improved. This is why I marked it as "major revision".

Reviewer 1 ·

Basic reporting

The manuscript provides sufficient information in introduction section. However, there are few suggestions for the improvement of the manuscript.

I suggest the author to ask a native English speaker to revise the manuscript for spelling and grammar mistakes. The author needs to carefully recheck the manuscript, and the missing references should be added. Additionally, author should add the latest references.

Regarding title of the manuscript, I recommend the author to change it. Title should clearly represent the inside message.

Experimental design

The study is looking original with well-defined research questions. It fills a knowledge gap of isolation, identification, and health hazards of beach bacteria. It comprehensively and rigorously describes the pathological effects and underlying mechanisms of this strain.
However, the manuscript lacks details in many methods used, which seems difficult to be repeated by other reserachers.
How did the author control fungal growth? Because fungal spores can easily grow on growth medium.
It is not clear whether each experiment was independently repeated three times, or the experiments were conducted once using three test samples.Please clarify it.

Validity of the findings

Findings in the present manuscript are looking good. Figures should be saved in high resolution format, and figure legends should be improved by adding sufficient information.

Additional comments

Results section should be written scientifically including the aim and outcomes of the experiments. If possible, add some references in results section too.

Reviewer 2 ·

Basic reporting

The aims and approach presented in the manuscript, entitled “Isolation, Identification, and Pathological Effects of Beach Sand Bacteria on Human Skin Keratinocytes” represent a relevant perspective for human skin infection on the beaches. The manuscript overall well written.

Experimental design

Overall the experimental design is good enough

Validity of the findings

Majority of the findings are novel and there is sufficient novelty in the current manuscript.

Additional comments

However, the manuscript has some important revision that need to be addressed.
I suggest minor revision
1. (Line 56-60) The following sentence should be changed with suitable one. “For example, Heaney and colleagues have reported that, following visits to both freshwater and marine beaches in different parts of the USA in 2003-2005, the incidence of diarrhea, vomiting, and nausea increased in visitors.
2. Figure 5. If possible mentioned the IC50 valve in the results.
3. (Line 61) The skin is the largest organ in body and provides the primary barrier against bacterial infections. Change to …..primary barrier against various infectious agents.
4. (Line 98-100) Change, the soil samples… to sand samples were collected from the seaside area.
5. (Line 103) Kang et al., 2015 change to italic Kang et al., 2015
6. (Line 131) 100 mg/ mL streptomycin change to 100 mg/mL streptomycin
7. (Line 206-208) An understanding of the pathological microbiome is necessary to understand microbes pathological to humans, and to enable the development of successful therapeutic approaches for their cure. Change this sentence to “An understanding of the human pathological microbiome is necessary to enable the development of successful therapeutic approaches for their cure”.
8. (Line 235) Change the “P. stutzeri” to’ FSRS”
9. Figures should be saved in high quality.
10. The legends should be improved for better understanding.

Annotated reviews are not available for download in order to protect the identity of reviewers who chose to remain anonymous.

Reviewer 3 ·

Basic reporting

The English of the manuscript is enough clear throughout the text.

Experimental design

The research is with in the scope of the journal and method adopted for investigation is technical and scientific.

Validity of the findings

conclusion is supporting the results and hypothesis of the manuscript.

Additional comments

This is an interesting work; the authors warn the people visiting beaches for luxury purposes to be harmful. Although there are interesting data presented in the manuscript, the article would need minor revisions before it could be accepted for publication. The authors may need to address the following suggestions for improving the quality of the paper

1. The authors did not mention the lethal dose or the IC50 concentration of the extract tested in cell proliferation data
2. (Line 57) remove “following” and visit to “visitors”
3. (Line 58) change “There have also been” to “There are several other similar reports”
4. Figure 2-1: cluster should be defined
5. Figure 4: IC50 value should be mentioned in the result if possible percentage of cell proliferation”
6. (Line 244) remove “as” after could
7. Please make sure that all the references are in the format of the journal.

---

## Round 0.2 · accepted · Accept

Your revised paper has been critically examined by two of the original reviewers. Both were satisfied with the changes you made.

Reviewer 1 ·

Basic reporting

Well explained; having brief background.

Experimental design

Novel idea, with good outcomes.

Validity of the findings

Authentic.

Reviewer 2 ·

Basic reporting

The manuscript was revised appropriately and I would recommend for publication in PeerJ.

Experimental design

The Experimental part was designed properly and written in a proper way.

Validity of the findings

The manuscript was revised appropriately and I would recommend for publication in PeerJ.

Additional comments

The manuscript was revised appropriately and I would recommend for publication in PeerJ.